# Stable reference gene selection for *Ophiocordyceps sinensis* gene expression studies under different developmental stages and light-induced conditions

Chaoqun Tong[1], Junhong Wei[1], Xianbing Mao[2], Guoqing Pan[1], Chunfeng Li[1]*, Zeyang Zhou[1,3]

**1** State Key Laboratory of Silkworm Genome Biology, Chongqing Key Laboratory of Microsporidia Infection and Prevention, Southwest University, Chongqing, China, **2** Chongqing Xinstant Biotechnology Co., Ltd., Chongqing, China, **3** College of Life Sciences, Chongqing Normal University, Chongqing, China

\* licf@swu.edu.cn

**Data Availability Statement:** All relevant data are within the paper and its Supporting Information files.

## Abstract

The molecular mechanism of Chinese cordyceps formation has received a substantial amount of attention because of its usage as traditional Chinese medicine. The formation process of Chinese cordyceps includes two parts: asexual proliferation (*Ophiocordyceps sinensis* proliferates in the hemolymph of *Thitarodes armoricanus* larvae) and sexual development (formation and development of fruiting bodies). Therefore, validation of reference genes under different development stages and experimental conditions is crucial for RT-qPCR analysis. However, there is no report on stable reference genes at the development stage of *O. sinensis* fruiting body. In this study, 10 candidate reference genes, *Actin*, *Cox5*, *Tef1*, *Ubi*, *18s*, *Gpd*, *Rpb1*, *Try*, *Tub1* and *Tub2*, were selected and calculated their expression stability using four methods: geNorm, NormFinder, BestKeeper, and Comparative $\triangle C_t$. After comprehensive analysis of the results of these four methods with RefFinder, we determined that the most stable reference genes during asexual reproduction of *O. sinensis* were *Tef1* and *Tub1*, while the most stable reference genes during fruiting body development were *Tyr* and *Cox5*, and the most stable reference genes under light-induced conditions were *Tyr* and *Tef1*. Our study provides a guidance for reference genes selections at different proliferation processes with light stress of *O. sinensis*, and represents a foundation for studying the molecular mechanism of Chinese cordyceps formation.

## Introduction

Chinese cordyceps, also named "Dong Chong Xia Cao" is a traditional Chinese medicine. It is an insect–fungus complex formed by an entomopathogenic fungus, *Ophiocordyceps sinensis*, that parasitizes the larvae of the insect *Thitarodes armoricanus*. Increasing in-depth research has revealed that Chinese cordyceps shows many pharmacological anti-oxidative [1], anti-inflammatory [2], hypoglycemic [3], and immunomodulatory [4] actions. Chinese cordyceps is endemic to the Qinghai–Tibet Plateau and mainly occurs at altitudes between 3000–5000 m.

**Funding:** Fundamental Research Funds for the Central Universities to Southwest University Award Number: XDJK2018AA001 Recipient: Guoqing Pan.

**Competing interests:** The authors have declared that no competing interests exist.

Extreme growth environments limit wild Chinese cordyceps yields. Therefore, researchers have begun to study artificial cultivation of Chinese cordyceps in low altitude areas and have made many breakthroughs [5, 6]. In the low-altitude laboratory, the researchers were able to complete the inoculation of *O. sinensis* in the larval stage of the *T. armoricanus*, and induce larval death and mummification to grow fruiting bodies [7]. However, low mummification rates and inconsistent fruiting body development are still major problems that affect artificial cultivation [8]. Consequently, many researchers have focused on fruiting body formation [7, 9] and the influence of environmental factors (such as temperature and light) on fruiting body development [10, 11].

Through transcriptome and proteome analyses, many differentially expressed genes were identified in different stages of fruiting body development [12]. These genes are involved in MAPK signal transduction, glucose metabolism [13], amino acid metabolism [14], lipid metabolism, and other pathways [15, 16]. Simultaneously, it was also found that *O. sinensis* is sensitive to light [11], and the mycelia synthesize melanin under light conditions (especially under UV irradiation) [17]. A large amount of omics data provided many new targets for *O. sinensis* artificial cultivation. However, no stable reference gene has been reported, which is very important for studying the molecular mechanism. Studying the key genes and molecular mechanisms in the process of *O. sinensis* infection, asexual proliferation, sexual development, and synthesis of active substances will become the research focus of artificial cultivation of Chinese cordyceps. During the formation of Chinese cordyceps, we usually focus on the formation of fruiting bodies and the effect of light on *O. sinensis*.

Reverse transcription quantitative PCR (RT-qPCR) is a widely used and authoritative method to evaluate gene expression. However, some variable factors may affect the accurate quantification of gene expression [18, 19], including the differences in materials, RNA extraction process, reverse transcription efficiency, and primer design. Therefore, stably expressed reference gene is needed to correct experimental errors [20]. It is generally believed that housekeeping genes are stably expressed and used for standardized correction of target genes. However, many studies have proved that the expression of common housekeeping genes will change due to different species and experimental treatments, and there were no konwn stable reference gene applicable to all conditions. Thus selections of a collection of reference genes is necessary for particular experimental models. The formation process of Chinese cordyceps involves the asexual and sexual multiplication stage of *O. sinensis*. In the stage of asexual proliferation, *O. sinensis* proliferates in the host hemolymph, and there will be morphological transformation between blastospores and hyphae [12]. In the stage of sexual development (fruiting bodies formation and maturity), similar to other edible fungi, light is usually used to regulate the phenotype of fruiting bodies. In this process, there is no report on the stable reference gene of *O. sinensis*.

In order to fill this gap, we explored the conditions of liquid fermentation and fruit-body induction, and observed and prepared *O. sinensis* samples at all stages of development as comprehensively as possible. Liquid culture were used to simulate the process of asexual proliferation, and through this method, blastospores and hyphae (or mycelial pellets) of *O. sinensis* under different light conditions were collected. In the process of sexual reproduction, we prepared a batch of *T. armoricanus* larvae infected with *O. sinensis*. Under the condition of artificial induction, the hyphal bodies in hemolymph and the fruiting bodies at different stages of development (form formation to maturity) were collected. By using the methods of geNorm [21], NormFinder [22], BestKeeper [23], and Comparative $\triangle C_t$ methods [24], the expression stabilities of ten common reference genes was analyzed.We screened the stable reference genes of *O. sinensis* in the asexual proliferation process, the fruiting body development process and

the light-induced process. Our results will provide guidelines for reference gene selection of RT-qPCR in *O. sinensis* gene expression analysis

## Materials and methods

### Strains cultivation and sample collection

For the asexual samples, *O. sinensis* was activated and cultivated with OS1 liquid medium (20% potato infusion, 2% glucose, 1% yeast extract, 0.5% peptone, 0.1% $KH_2PO_4$, 0.025% $MgSO_4$), and shaken at 16°C and 100 rpm for 40 d. The blastospore suspension was collected as seed solution with lens wiping paper, and inoculated in OS1 and 1/4 SDY liquid medium (1% glucose, 0.5% yeast extract, 0.25% peptone), and then shaken at 16°C and 120 rpm. Light and dark treatments were performed on the samples cultivated with OS1 liquid medium. The blastospore and mycelium samples were harvested in 1/4 SDY liquid medium at 30 d and 45 d, respectively. The loose and dense mycelium balls were harvested in OS1 liquid medium at 30 d and 70 d, respectively.

The larvae infected with *O. sinensis* and the fruiting bodies were reared in a low-altitude laboratory (Jiangjin, Chongqing). The hyphal body samples were collected from the hemolymph of larvae inoculated with *O. sinensis*. Mummified larvae were buried in culture medium for moisture retention during the fruiting body formation and development. Light (light: dark = 12 h: 12 h) and dark (24h dark) treatments were performed on the samples after fruiting body formation.

### RNA isolation and cDNA synthesis

The samples were ground in liquid nitrogen, and 200 mg samples were used to isolate total RNA with the Total RNA Kit II (R6934-02, Omega, Norcross, Georgia USA). Then, 2 μg total RNA of each sample was used for reverse transcription synthesis of cDNA with Hifair® Ⅲ 1st Strand cDNA Synthesis SuperMix for qPCR (YEASEN, Shanghai, China) and stored at −20°C after 10-fold dilution. qPCR primers were designed by the NCBI Primer-BLAST tool (https://www.ncbi.nlm.nih.gov/tools/primer-blast/).

### Candidate reference gene selection and primer design

According to similar studies in other fungi [18, 19, 25, 26], ten housekeeping genes were selected as candidate reference gene targets, including actin protein (*Actin*), cytochrome c oxidase polypeptide V (*Cox5*), translation elongation factor EF-1 alpha (*Tef1*), polyubiquitin binding protein (*Ubi*), 18s ribosomal (*18s*), glyceraldehyde-3-phosphate dehydrogenase (*Gpd*), polymerase II large subunit (*Rpb1*), tryptophan synthase (*Try*), and tubulin beta chain (*Tub1*, *Tub2*); the tubulin beta chain has two copies in *O. sinensis* that we named *Tub1* and *Tub2*. The candidate genes covered multiple processes such as transcription and translation (*Rpb1* and *Tef1*), protein synthesis and degradation (*18s* and *Ubi*), cytoskeletal structure (*Actin*, *Tub1*, and *Tub2*), and metabolism (*Cox5*, *Gpd*, and *Try*). Details of reference genes and primers are shown in Table 1. Specificity of primer pairs for each candidate gene was confirmed using melting curve analysis (S1 Fig).

### RT-qPCR

The samples were divided into three groups: asexual proliferation (all liquid fermentation samples), sexual proliferation (samples during fruiting body development), and light treatment (light- and dark-treated samples during liquid fermentation and fruiting body development). Quantitative PCR was used to detect the expression of ten reference genes in each sample with

**Table 1. Details of primers for RT-qPCR.**

| Gene name | Accession number [1] | Primer Sequence (5'-3') [2] | Amplicon length(bp) | Tm (˚C) | Efficiency (%) |
|---|---|---|---|---|---|
| *Actin* | KAF4512910.1 | F:TGATGATGCTCCCCGAGCTG | 155 | 64.0 | 97 |
| | | R:CACCGTGCTCGATGGGGTAT | | 62.8 | |
| *Cox5* | KAF4508291.1 | F:GTCGACCCATGCCATCTCCA | 100 | 63.5 | 94 |
| | | R:ACGCAGGGACATCCAAAGGT | | 61.5 | |
| *Tef1* | KAF4504225.1 | F:GCGTTGAGGCTTTCACCGAC | 105 | 61.9 | 105 |
| | | R:CAGCAGCCTTCTCGACAGAC | | 59.1 | |
| *Ubi* | KAF4508947.1 | F:AGATGGCCTCGTTGTCTCGG | 147 | 62.2 | 93 |
| | | R:AGAAAAGTGCCGTTGGGGGA | | 64.0 | |
| *18s* | KC184161.1 | F:AGATTCAAAGCCAATGCCCC | 129 | 61.4 | 108 |
| | | R:TGGCCACTACCCAAACATCG | | 61.5 | |
| *Gpd* | KAF4510795.1 | F:TCTTCACCACCACGGACAAG | 143 | 59.7 | 97 |
| | | R:GGAAATGACATCGGCCTTGC | | 61.6 | |
| *Rpb1* | KAF4510254.1 | F:CAAGCGTGTCGACTTTTCGG | 113 | 61.6 | 94 |
| | | R:CAGTCTCGGGATACGTCAGC | | 59.8 | |
| *Try* | KAF4507068.1 | F:GAAGGCGCAGATGATGGAGA | 106 | 60.8 | 104 |
| | | R:GTCGTTGGCAAAGGGGTAGA | | 60.0 | |
| *Tub1* | KAF4510680.1 | F:ATGTCGTCCACCTTTGTCGG | 151 | 60.3 | 100 |
| | | R:ACTCGGCTTCGGTAAACTCC | | 59.6 | |
| *Tub2* | KAF4504885.1 | F:TATCCAGACAGCCCTTTGCG | 234 | 61.1 | 101 |
| | | R:TGGTACTGCTGGTACTCGGA | | 59.6 | |

[1] The accession number comes from NCBI.

[2] F, forward primer; R, reverse primer.

LightCycler® 96 (Roche, Indianapolis, IN, USA). The PCR mix (10 μL) contained 5 μL Hieff® qPCR SYBR Green Master Mix (YEASEN, Shanghai, China), 0.5 μL per primer, 1 μL 10-fold diluted cDNA, and 3 μL ddH$_2$O. The PCR conditions were as follows: 95˚C for 5 min, followed by 40 cycles of 95˚C for 10 s and 60˚C for 30 s. After the final amplification cycle, the melting curve analysis was performed as follows: 65˚C for 60 s, 65–95˚C with 0.5˚C increments, and 97˚C for 10 s.

## Statistical analysis

The expression stability of ten candidate reference genes was analyzed using geNorm, Norm-Finder, BestKeeper, and Comparative $\triangle C_t$ methods. GeNorm program converts cycle number (*Ct*) into relative gene expression ($2^{\triangle Ct}$), evaluated by calculating the stable value (M) of candidate reference gene expression, and finally, two or more reference gene combinations were provided by the analysis results of pairwise variation (Vn/Vn+1) [21]. NormFinder is also sorts the stability of reference genes according to M, and only one stable reference gene can be provided [22]. BestKeeper evaluated the stability of reference gene expression according to standard deviation (SD) and coefficient of variation (CV), the more stable genes have lower SD values [23].$\triangle C_t$ method is to compare the $\triangle Ct$ values of paired genes in different samples. If the $\triangle Ct$ values of these two genes remain unchanged in all samples, it is considered that these two genes are stably expressed in these samples; Otherwise, the expression of one or two genes is considered unstable. The final gene stability is sorted according to the change of $\triangle C_t$ value [24]. The overall final ranking was calculated by RefFinder(http://blooge.cn/RefFinder/) [27], which assigns the appropriate weights to individual genes based on the ranking of each

program and calculates the geometric mean of their weights for the overall final ranking. The data analysis was performed using LightCycler® 96 Software 1.1 (Roche), and visualization using Origin. Each of the above experiments was independently repeated at least 3 times.

## Results

### Sample collection

Through liquid fermentation, we collected *O. sinensis* samples with five morphologies (Fig 1A). They are blastospores, hyphae, loose hyphal spheres, and compact hyphal spheres under light

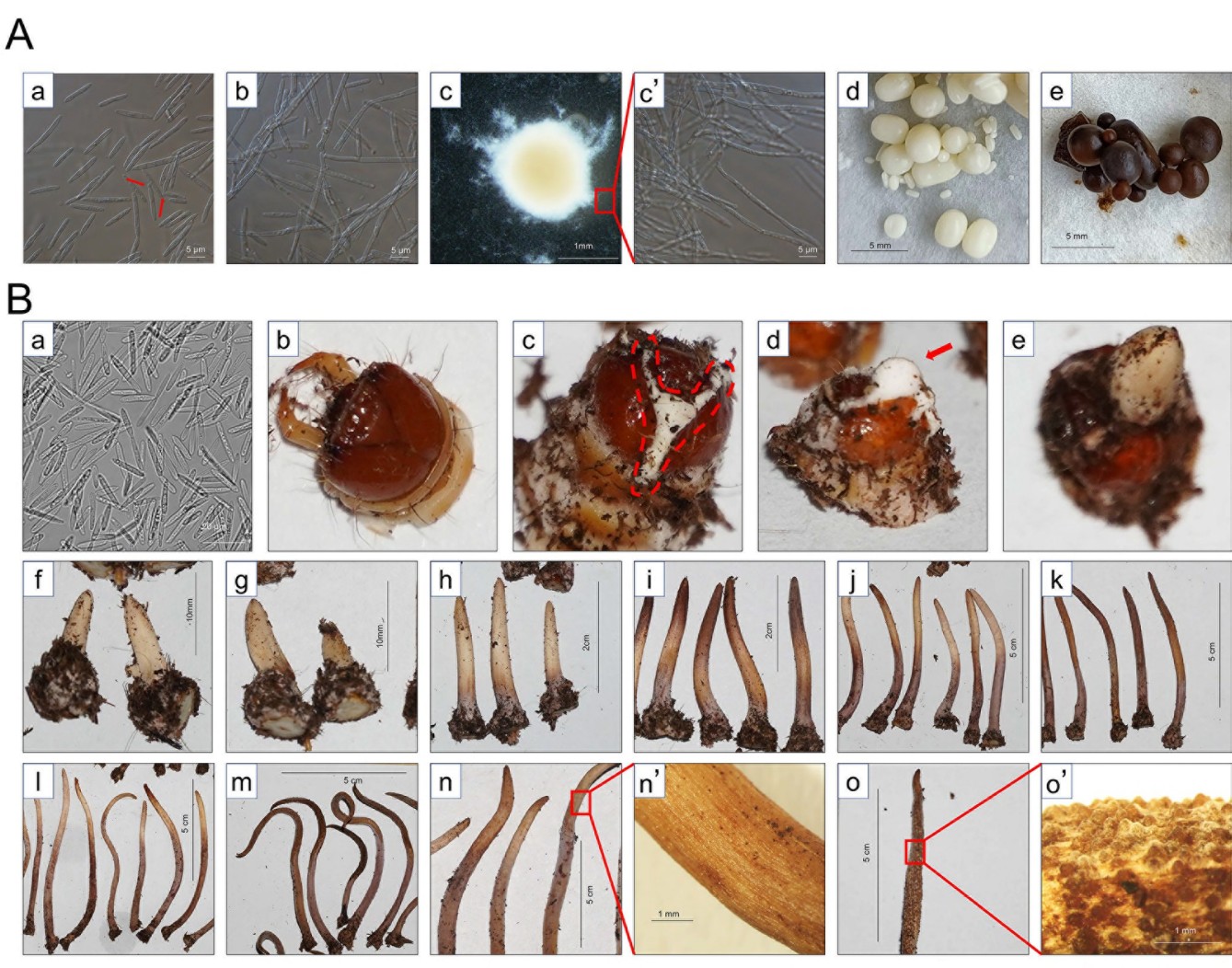

**Fig 1. Samples of *O. sinensis* under different proliferation conditions.** (**A**) Morphological observation of *O. sinensis* in 1/4 SDY liquid medium, in the early stage of fermentation, fusiform blastospores proliferated by budding (A-a red arrows), after cultivation for 45 d, most of the blastospores germinated into hyphae (A-b). Fermentation culture of *O. sinensis* in OS1 liquid medium, the hyphae aggregated into mycelial pellets (A-c), after culturing for 70 days, the mycelial pellets became compact and smooth, which were white under dark conditions (A-d) and black under light conditions (A-e). (**B**) Morphological observation of *O. sinensis* at different stages during infecting *T. armoricanus*. (B-a) *O. sinensis* proliferates in *T. armoricanus* larvae hemolymph in the form of hyphal body. The infected larvae died and mummified in 1 month (B-b), and the fungus penetrated the larval epidermis at the head shell and form a 'Y-type' hyphal structure (B-c). The 'Y-type' hyphal structure swelled and bulged (B-d), and finally developed into the primordium (B-e). After primordium formation, mummified larvae were treated in the dark or light environment, respectively. And the samples were taken and photographed at 7 d (dark: Fig 1B-f, light: Fig 1B-g), 18 d (dark: Fig 1B-h, light: Fig 1B-i), 32 d (dark: Fig 1B-j, light: Fig 1B-k), 47 d (dark: Fig 1B-l, light: Fig 1B-m), and 62 d (dark: Fig 1B -n, light: Fig 1B-o) after dark or light treatment, respectively.

and dark conditions. In the early stage of fermentation, 5–10 μm fusiform blastospores were proliferated by budding (Fig 1A-a red arrows). After continued cultivation for 45 d, most of the blastospores germinated into hyphae (Fig 1A-b). In OS1 medium, the hyphae aggregated into mycelial pellets (Fig 1A-c). The mycelial pellets become dense and smooth after 70 days of culture, and appear white (Fig 1A-d) and black (Fig 1A-e) respectively under the induction of dark and light conditions.

The hyphal body samples were harvested from larval hemolymph (Fig 1B-a) and were observed as fusiform-like blastospores. The hyphal body length (10–15 μm) was longer than that of the blastospores (5–10 μm). The mummified larvae were embedded in the culture medium to induce fruiting body formation. Initially, *O. sinensis* only proliferates in the host, and the mycelium does not penetrate the host epidermis so that the head shell of the mummified larvae (Fig 1B-b) did not change significantly. However, the fungus preferentially penetrated the larval epidermis at the head shell with proliferation and form a 'Y-type' hyphal structure (Fig 1B-c). Subsequently, the 'Y-type' hyphal structure swelled and bulged (Fig 1B-d), and finally developed into the primordium (Fig 1B-e). The results of the mummified larvae when primordia exposed to light and dark treatments showed that there was no significant difference in fruiting body phenotypes between dark (Fig 1B-f) and light (Fig 1B-g) treatments in the early stage of fruiting body development. When the fruiting body developed to approximately 2 cm, the color of the top of the fruiting body became yellow-brown with light treatment (Fig 1B-i) but was still white under dark treatment. Upon continued cultivation of the fruiting body to 7 cm, the color of fruiting body was yellow-brown under light treatment (Fig 1B-k), whereas the color only got darker at the top and base under dark treatment (Fig 1B-j). After 40 days of fruiting body development, the middle and upper part of the light-treated fruiting bodies formed bulges (Fig 1B-m) and finally developed into asci (Fig 1B-o), whereas the dark-treated fruiting bodies still had a smooth surface (Fig 1B-i, n).

## Expression profiling of candidate genes

The expression levels of 10 genes were calculated by quantification cycles and shown with box plots (Fig 2). The samples were divided into three groups (asexual proliferation, sexual proliferation, and light treatment), and the candidate genes had the same expression trend in different groups. Except for the Cq value of *18s* distributed around 10, the expression levels of other genes were in the range of 20–30 indicating that the expression level of *18s* was significantly higher than that of other genes. In the sexual proliferation and light treatment groups,

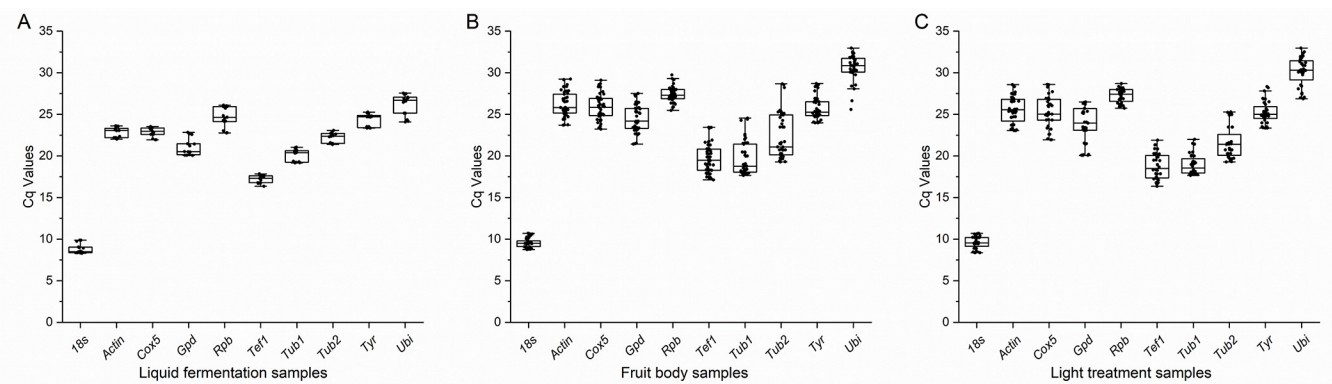

**Fig 2. Candidate gene expression levels under different grouping conditions.** (A) Asexual proliferation samples. (B) Fruiting body development samples. (C) Light stress samples samples.

the Cq value of *Ubi* reached 32.97, which indicated that the expression of *Ubi* was very low in the samples.

## geNorm analysis

The average expression stability (M value) of the ten candidates was calculated with the GENORM software. The M value is defined as the average pairwise variation of a particular gene with all other reference genes within a given group of cDNA samples. where a low value represents stable and a high value represents unstable expression. A low stability value (M) indicates high expression stability and the geNorm software defaults that M = 1.5 is the threshold of gene stability. According to the calculation results, 7 genes in asexual proliferation samples were within the acceptable range (Table 2), and the most stable reference genes are *Tub1* / *Tyr* (M = 0.153). Similarly, there are 6 genes within the acceptable range in the fruit body development samples (Table 3) and the light stress samples (Table 4), and the most stable reference genes are *Cox5*/ *Gpd* (M = 0.619) and *Tyr* / *Tef1* (M = 0.579) respectively.

Further, the minimal number of genes for qRT-PCR normalization were determine by estimated pairwise variation (Vn/Vn+1) in geNorm. It is generally considered that Vn/Vn +1 < 0.15 is the cut-off threshold, and no additional reference gene is required. Therefore, no more than 2 reference genes are required in ll three groups of samples (0.007 in asexual proliferation samples, 0.012 in fruit body development samples and 0.011 in light stress samples) to normalization the qRT-PCR results (Fig 3).

## NormFinder analysis

NormFinder calculate the stability based on an ANOVA-based algorithm, and also considers the best two genes with minimal combined intra- and intergroup expression variation. In

**Table 2. Stability analysis of candidate reference genes in asexual proliferation samples.**

| Gene | geNorm | Comparative $\triangle C_t$ | NormFinder | BestKeeper [4] | | Comprehensive Stability value [5] | Ranking [6] |
|---|---|---|---|---|---|---|---|
| | Stability value [1] | Average of STDEV [2] | Stability value [3] | SD | CV | | |
| *Tef1* | 0.317 | 2.51 | 0.15 | 0.39 | 2.25 | 1.86 | 1 |
| *Tub1* | 0.153 | 2.52 | 0.077 | 0.67 | 3.33 | 1.97 | 2 |
| *Tub2* | 0.256 | 2.52 | 0.141 | 0.56 | 2.5 | 2.63 | 3 |
| *Tyr* | 0.153 | 2.54 | 0.184 | 0.71 | 2.91 | 3.13 | 4 |
| *18s* | 0.457 | 2.55 | 0.304 | 0.48 | 5.48 | 4.16 | 5 |
| *Actin* | 0.638 | 2.69 | 0.255 | 0.54 | 2.37 | 4.82 | 6 |
| *Rpb1* | 0.913 | 3.15 | 1.427 | 0.99 | 4.02 | 7 | 7 |
| *Gpd* | 2.05 | 6.18 | 5.527 | 2.62 | 13.33 | 8 | 8 |
| *Cox5* | 2.962 | 6.54 | 5.952 | 2.85 | 13.33 | 9 | 9 |
| *Ubi* | 3.87 | 7.5 | 7.072 | 3.34 | 13.68 | 10 | 10 |

[1] The stability value calculated by geNorm is the standard deviation of the expression variation of the two genes. A low stability value indicates high expression stability.

[2] Comparative $\Delta C_t$ method ranks candidate gene stability according to the repeatability of gene expression differences. A low Average of STDEV indicates high expression stability.

[3] The stability value calculated by NormFinder. A high stability value indicates low expression stability.

[4] BestKeeper analyzes expression stability by calculating cycle threshold (Ct) data variation. A low SD (standard deviation) and CV (Coefficient of Variance) value indicates high expression stability.

[5] Comprehensive stability value assigns the appropriate weights to individual genes based on the ranking of each program and calculates the geometric mean of their weights for the overall final ranking.

[6] Ranking assigned to each gene according to its comprehensive stability value.

**Table 3. Stability analysis of candidate reference genes in fruiting body development samples.**

| Gene | geNorm | Comparative $\triangle C_t$ | NormFinder | BestKeeper [4] | | Comprehensive Stability value [5] | Ranking [6] |
|---|---|---|---|---|---|---|---|
| | Stability value [1] | Average of STDEV [2] | Stability value [3] | SD | CV | | |
| *Tyr* | 0.689 | 1.42 | 0.307 | 1.09 | 4.22 | 1.86 | 1 |
| *Cox5* | 0.619 | 1.46 | 0.307 | 1.29 | 5 | 2.11 | 2 |
| *Gpd* | 0.619 | 1.5 | 0.394 | 1.38 | 5.62 | 3.13 | 3 |
| *Actin* | 0.658 | 1.5 | 0.31 | 1.28 | 4.9 | 3.22 | 4 |
| *18s* | 1.031 | 2.04 | 1.462 | 0.4 | 4.18 | 4.45 | 5 |
| *Rpb1* | 0.867 | 1.65 | 0.739 | 0.67 | 2.45 | 4.56 | 6 |
| *Tef1* | 0.757 | 1.53 | 0.717 | 1.43 | 7.28 | 5.44 | 7 |
| *Tub1* | 1.17 | 1.96 | 1.518 | 1.7 | 8.66 | 7.74 | 8 |
| *Tub2* | 1.302 | 2.17 | 1.789 | 2.34 | 10.48 | 9.24 | 9 |
| *Ubi* | 2.003 | 4.81 | 4.718 | 1.91 | 6.36 | 9.74 | 10 |

The footer is the same as Table 2

asexual proliferation samples, the order of stability (most stable to least stable) is *Tub1 > Tub2 > Tef1 > Tyr > Actin > 18s > Rpb > Gpd > Cox5 > Ubi* (Table 2). In fruiting body development samples, the order of stability (most stable to least stable) is *Tyr > Cox5 > Actin > Gpd > Tef1 > Rpb > 18s > Tub1 > Tub2 > Ubi* (Table 3). In light stress samples, the order of stability (most stable to least stable) is *Tef1 > Tyr > Actin > Ubi > Rpb > 18s > Tub1 > Tub2 > Gpd > Cox5* (Table 4). In the calculation results of NormFinder and geNorm, the most and least stable reference genes are basically the same.

## BestKeeper analysis

BestKeeper evaluates the stability of reference genes by calculating the standard deviation (SD) and the coefficient of variation (CV) of Cq values. It is generally believed that the reference genes with SD < 1 are stable, and the smaller the CV indicates the higher stability. As the result, 7 genes (*Tef1*, *Actin*, *Tub2*, *Tyr*, *Tub1*, *Rpb*, *18s*) in asexual proliferation samples (Table 2), 2 genes (*Rpb* and *18s*) in fruiting body development samples (Table 3) and 3 genes (*Rpb*, *Tub1*, *18s*) in light stress samples (Table 4) are stable. The results of BestKeeper are different from NormFinder and geNorm, so it is necessary to evaluate the results of multiple software.

**Table 4. Stability analysis of candidate reference genes in light stress samples.**

| Gene | geNorm | Comparative $\triangle C_t$ | NormFinder | BestKeeper [4] | | Comprehensive Stability value [5] | Ranking [6] |
|---|---|---|---|---|---|---|---|
| | Stability value [1] | Average of STDEV [2] | Stability value [3] | SD | CV | | |
| *Tyr* | 0.579 | 1.59 | 0.289 | 1 | 3.99 | 1.68 | 1 |
| *Tef1* | 0.579 | 1.62 | 0.289 | 1.4 | 7.37 | 1.93 | 2 |
| *Actin* | 0.709 | 1.69 | 0.322 | 1.4 | 5.35 | 3.57 | 3 |
| *Rpb1* | 0.843 | 1.7 | 0.766 | 0.7 | 2.59 | 3.76 | 4 |
| *18s* | 0.924 | 1.91 | 1.172 | 0.5 | 5.43 | 3.83 | 5 |
| *Ubi* | 0.755 | 1.75 | 0.336 | 1.3 | 4.42 | 4.47 | 6 |
| *Tub1* | 1.061 | 2.05 | 1.466 | 1 | 5.19 | 5.66 | 7 |
| *Tub2* | 1.172 | 2.15 | 1.536 | 1.5 | 6.87 | 8 | 8 |
| *Gpd* | 1.793 | 4.1 | 3.813 | 2.4 | 10.4 | 9.24 | 9 |
| *Cox5* | 2.279 | 4.22 | 3.958 | 2.3 | 9.17 | 9.74 | 10 |

The footer is the same as Table 2

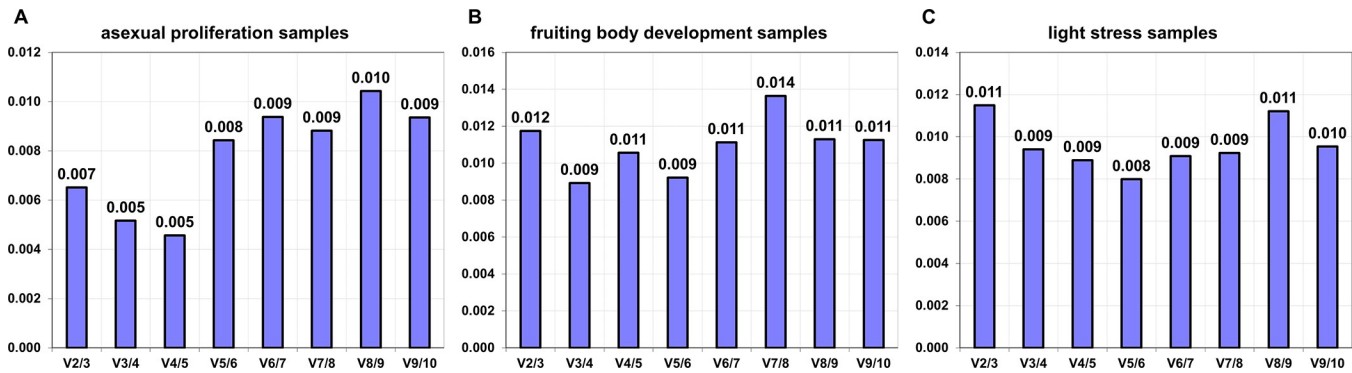

**Fig 3. Pairwise variation (V) of 10 reference genes calculated by geNorm.** (A) Asexual proliferation samples. (B) Fruiting body development samples. (C) Light stress samples samples.

## Comparative Δ*Ct* analysis

The comparative Δ*Ct* method evaluate reference gene stability by the repeatability of gene expression differences. In asexual proliferation samples (Table 2), the results of Comparative Δ*Ct*, NormFinder and geNorm are very similar. The four most stable reference genes are *Tef1*, *Tub1*, *Tub2* and *Tyr*. Similarly, the same stability trend also appeared in fruiting body development samples (Table 3) and light stress samples samples (Table 4).

## Comprehensive ranking order

The reference genes stability calculated by different methods is slightly different, so RefFinder was used to comprehensively calculate the overall final ranking. RefFinder assigns the appropriate weights to individual genes based on the ranking of each program and calculates the geometric mean of their weights for the overall final ranking. we recommended that the most stable reference genes during asexual reproduction of *O. sinensis* are *Tef1* and *Tub1* (Table 2), the most stable reference genes during fruiting body development were *Tyr* and *Cox5* (Table 3), and the most stable reference genes under light-induced conditions were *Tyr* and *Tef1* (Table 4).

## Discussion

RT-qPCR is an important method for studying gene expression that can help eliminate experimental errors, and selecting a stably expressed reference gene for RT-qPCR analysis is necessary [26]. In this study, the asexual proliferation process, fruiting body development process, and effects of light on *O. sinensis* were recorded in detail. Additionally, we assessed the stability of 10 candidate reference genes under three different experimental conditions. These results provide a reference for studying the gene expression of *O. sinensis* under different conditions. Through this study, we found that the reference genes stably expressed by *O. sinensis* are not the same under different proliferation forms and treatment conditions, which further demonstrated the importance of selecting appropriate reference genes.

The formation process of Chinese cordyceps can be divided into two stages. One is the proliferation stage of *O. sinensis* in the hemolymph of *T. armoricanus* larvae. It is generally believed that, in this stage, *O. sinensis* mainly propagates asexually in the form of hyphal bodies [28]. At the subsequent stage, the hyphal bodies are induced by certain signal substances to transform by hyphae proliferation [29]. Our results revealed that, as the content of *O. sinensis* mycelium in the host hemolymph increased, the mycelium began clustering and caused the host larvae died and mummified. The second stage is the formation of fruiting bodies [12]. It

is generally believed that, similar to mushrooms [30], the formation of fruiting bodies of *O. sinensis* is also the result of sexual mating, and asci will be formed in the middle parts of the fruiting bodies at the later stage of development [12]. Although the molecular mechanism of Chinese cordyceps formation has not been completely analyzed, the dimorphic transition (from blastospores or hyphal bodies to mycelial pellets) [31] and the developmental regulation of fruiting bodies have attracted the attention of many scholars.

In this study, the asexual proliferation process of *O. sinensis* was simulated by liquid fermentation. The formation of blastospores (similar to hyphal bodies), mycelia, and mycelial pellets was more accurately reproduced. Using different types of samples, we screened the stably expressed reference genes *Tef1* and *Tub1* during asexual proliferation. Although it could not be proven that the proliferation process of *O. sinensis* in liquid fermentation and host hemolymph was the same, the similar morphological changes can also provide a reference for studying the dimorphic transition of *O. sinensis* [29]. Our findings provide a tool for future research on morphogenesis and morphological changes of *O. sinensis*.

The formation and development of fungal fruiting bodies involve complex gene regulatory networks, and there are great differences among different species. Therefore, different experimental models need to re-evaluate the stability of internal reference genes. In *Cordyceps militaris*, the best reference genes for inducing fruiting bodies in wheat medium and pupae are *gpd*/*rpb1* and *rpb1*/*try*, respectively [32]. However, during the fruiting bodies development of *Flammulina velutipes*, *Botrytis cinerea* and *Agaricus bisporus*, the stable reference genes are *gapdh*, *tubulin* and *EF1-α*, respectively [33–35]. In the research related to the development of *O. sinensis* fruiting body, no research has evaluated the best reference genes yet. In this study, we collected samples from the mummified larva until the fruiting body matured and formed asci. By using fruiting body samples from various developmental stages, we determined that *Tyr* and *Cox5* represented stably expressed reference genes during the fruiting body development process.

During the process of asexual proliferation and fruiting body development, we found that light can affect *O. sinensis* color and ascus formation. Fungi can respond to light in various ways. For examples, fungi can respond to different wavelengths of light through photoreceptor proteins, and can also regulate the expression of downstream genes through light-induced intracellular ROS and NOS [36, 37]. In *C. militaris*, researchers usually use *Rbp1* as an internal reference gene when studying the regulation of light in fungal gene expression [32]. However, in this study, we found that *Rbp1* is not a stable reference gene for *O. sinensis* response to light. In a previous study, the researchers used *18s* as a reference gene to explore the expression pattern of the blue light receptor *OsWC1* in *O. Sinensis* [38]. It is generally believed that *18s* expression is stable and not easily affected by other conditions, but the expression level of *18s* is very high compared with other genes, and it is likely to exceed the confidence interval of the standard curve [39, 40]. In this study, we determined that the reference genes *Tyr* and *Tef1* are more stable than *18s* and *Rbp1*, which can be used for follow-up studies of *O. sinensis* responding to light.

## Conclusions

This study comprehensively described and collected samples of various phenotypes of sexual and asexual proliferation of *O. sinensis*, including blast spores, hyphae and hyphal spheres in liquid fermentation, hyphal body in the host hemolymph and fruiting bodies at various development stages. At the same time, the influence of light on each stage was studied. Moreover, three groups of samples (asexual proliferation, fruiting body formation and development, and response to light stress) have been screened for stable reference genes that are suitable for different experimental models through four methods (geNorm, Comparative $\triangle C_t$, NormFinder, and BestKeeper). This is the first time to screen reference genes comprehensively in *O. sinensis*.

This study provides a reliable tool for further studying the gene expressions in the process of *O. sinensis* morphological transformation, fruiting body development, and response to light stress.

## Supporting information

**S1 Fig. Melting curves and standard curve of candidate reference genes.** Melting curves were analyzed by LightCycler® 96 Software 1.1 (Roche). The standard curve was amplified from 10-fold diluted standard plasmid.
(TIF)

## Acknowledgments

We thank prof. Jialing Bao (State Key Laboratory of Silkworm Genome Biology, Southwest University) for valuable suggestions and editing the draft of this manuscript. We thank Liwen Bianji (https://new.liwenbianji.cn) for its linguistic assistance during the preparation of this manuscript.

## Author Contributions

**Conceptualization:** Chaoqun Tong, Guoqing Pan, Chunfeng Li, Zeyang Zhou.

**Methodology:** Xianbing Mao, Chunfeng Li.

**Software:** Chaoqun Tong.

**Validation:** Chaoqun Tong, Junhong Wei.

**Writing – original draft:** Chaoqun Tong.

**Writing – review & editing:** Chunfeng Li.

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
