## [Decision Letter · Decision Letter 0]

2 Feb 2023

PONE-D-22-31224Stable reference gene selection for Ophiocordyceps sinensis gene expression studies under different developmental stages and light-induced conditionsPLOS ONE

Dear Dr. li,

Thank you for submitting your manuscript to PLOS ONE. After careful consideration, we feel that it has merit but does not fully meet PLOS ONE’s publication criteria as it currently stands. Therefore, we invite you to submit a revised version of the manuscript that addresses the points raised during the review process.

The manuscript "Stable reference gene selection for Ophiocordyceps sinensis gene expression studies under different developmental stages and light-induced conditions" submitted by Chaoqun Tong et al is good work but it lack novelties. Similar kind of articles using Cordyceps sinensis and other species have been published in different journals. I can quote some of the articles here

1. SU Qiang-Jun, XIA Ying-Xia, XIE Fang, Uwitugabiye VESTINE, CHEN Zhao-He, ZHOU Gang. Screening of the reference genes for qRT-PCR analysis of gene expression in Ophiocordyceps sinensis[J].MYCOSYSTEMA, 2021, 40(7): 1712-1722.

2. Gangzheng Wang, Huijiao Cheng, Min Li, Chenghua Zhang, Wangqiu Deng, Taihui Li, Selection and validation of reliable reference genes for Tolypocladium guangdongense gene expression analysis under differentially developmental stages and temperature stresses,

Gene, Volume 734, 2020, 144380, ISSN 0378-1119, https://doi.org/10.1016/j.gene.2020.144380.

3. Tiantian Lian, Tao Yang, Guijun Liu, Junde Sun, Caihong Dong, Reliable reference gene selection for Cordyceps militaris gene expression studies under different developmental stages and media, FEMS Microbiology Letters, Volume 356, Issue 1, July 2014, Pages 97–104, https://doi.org/10.1111/1574-6968.12492

How will you justify the novelty of work when such work has been done by several authors on the same fungi.

The language correction is required through out the manuscript.

We look forward to receiving your revised manuscript.

Kind regards,

Shwet Kamal, Ph.D

Academic Editor

PLOS ONE

Journal Requirements:

“The authors received no specific funding for this work. No Funding”

Additional Editor Comments:

The manuscript "Stable reference gene selection for Ophiocordyceps sinensis gene expression studies under different developmental stages and light-induced conditions" submitted by Chaoqun Tong et al is good work but it lack novelties. Similar kind of articles using Cordyceps sinensis and other species have been published in different journals. I can quote some of the articles here

1. SU Qiang-Jun, XIA Ying-Xia, XIE Fang, Uwitugabiye VESTINE, CHEN Zhao-He, ZHOU Gang. Screening of the reference genes for qRT-PCR analysis of gene expression in Ophiocordyceps sinensis[J].MYCOSYSTEMA, 2021, 40(7): 1712-1722.

2. Gangzheng Wang, Huijiao Cheng, Min Li, Chenghua Zhang, Wangqiu Deng, Taihui Li, Selection and validation of reliable reference genes for Tolypocladium guangdongense gene expression analysis under differentially developmental stages and temperature stresses,

Gene, Volume 734, 2020, 144380, ISSN 0378-1119, https://doi.org/10.1016/j.gene.2020.144380.

3. Tiantian Lian, Tao Yang, Guijun Liu, Junde Sun, Caihong Dong, Reliable reference gene selection for Cordyceps militaris gene expression studies under different developmental stages and media, FEMS Microbiology Letters, Volume 356, Issue 1, July 2014, Pages 97–104, https://doi.org/10.1111/1574-6968.12492

How will you justify the novelty of work when such work has been done by several authors on the same fungi.

The language correction is required through out the manuscript.

Reviewers' comments:

Reviewer's Responses to Questions

**Comments to the Author**

1. Is the manuscript technically sound, and do the data support the conclusions?

Reviewer #1: Partly

Reviewer #2: Yes

Reviewer #3: Partly

2. Has the statistical analysis been performed appropriately and rigorously? 

Reviewer #1: Yes

Reviewer #2: Yes

Reviewer #3: Yes

3. Have the authors made all data underlying the findings in their manuscript fully available?

Reviewer #1: Yes

Reviewer #2: No

Reviewer #3: Yes

4. Is the manuscript presented in an intelligible fashion and written in standard English?

Reviewer #1: No

Reviewer #2: Yes

Reviewer #3: Yes

5. Review Comments to the Author

Reviewer #1: The present work related to reference gene expression in O. sinensis. Lian et al (2014) ref no 36 also performed the similar work. Here, authors need to perform the detail work in the different light intensity or different light interval (hours or days) for the expression analysis. Beside this manuscript also required proofreading.

Reviewer #2: The current study provides an overview of how stable reference genes might be chosen for Ophiocordyceps sinensis under different developmental and light induced conditions. The manuscript is well written and structured. However, the manuscript required several minor corrections before publication.

Comments:

1. Cite more relevant and recent literature under discussion section.

2. Reference no. 4, 13, 21, 24, 31, 32, 33: correct the references.

3. Line no 171, 341: write the scientific name in italics.

4. Line no. 218-219: avoid the repetitions.

5. Line no 382: write the name of genes in italics.

6. Table no 2, 3: check the alignment.

Reviewer #3: Required grammer check or language editing must improved. Imporve the quality of paper by following the comments attached in this link. Please ensure that you follow PLOS one guidelines for presenting name of authors/abbreviation of journals/year of publication in bold/volume journals in italics etc., in the refeerence section. and also in other sections.

6. PLOS authors have the option to publish the peer review history of their article (what does this mean?). If published, this will include your full peer review and any attached files.

Reviewer #1: No

Reviewer #2: **Yes: **Rakesh Kumar Bairwa

Reviewer #3: No

---

## [Author Response · Author response to Decision Letter 0]

3 Mar 2023

Dear Editor and Reviewers,

Thank you much for your kind and constructive comments which served as a welcome guideline to improve the manuscript. According to your suggestion, we have revised our manuscript carefully. We read the manuscript carefully and provide a point-by-point response to the reviewer’s comments. All revised portions are made blue background in the Response to Reviewers.

Sincerely, 

Chunfeng Li，PhD

State Key Laboratory of Silkworm Genome Biology, Southwest University, Chongqing, 400715, China.

licf @swu.edu.cn

---

## [Editor Report · Decision Letter 1]

3 Apr 2023

Stable reference gene selection for Ophiocordyceps sinensis gene expression studies under different developmental stages and light-induced conditions

PONE-D-22-31224R1

Dear Dr. li,

We’re pleased to inform you that your manuscript has been judged scientifically suitable for publication and will be formally accepted for publication once it meets all outstanding technical requirements.

Kind regards,

Shwet Kamal, Ph.D

Academic Editor

PLOS ONE
---

## [Editor Report · Acceptance letter]

12 Apr 2023

PONE-D-22-31224R1 

Stable reference gene selection for *Ophiocordyceps sinensis* gene expression studies under different developmental stages and light-induced conditions 

Dear Dr. Li :

I'm pleased to inform you that your manuscript has been deemed suitable for publication in PLOS ONE. Congratulations! Your manuscript is now with our production department. 

Kind regards, 

on behalf of

Dr. Shwet Kamal 

Academic Editor

PLOS ONE